# Deciphering Breast Tumor Heterogeneity Through Patient-Derived Organoids and Circulating Tumor Cells

**DOI:** 10.3390/jpm15070271

**Published:** 2025-06-25

**Authors:** Benedetta Policastro, Nikoline Nissen, Carla L. Alves

**Affiliations:** Cancer Research Unit, Department of Molecular Medicine, University of Southern Denmark, 5000 Odense, Denmark; policastro@health.sdu.dk (B.P.); niknissen@health.sdu.dk (N.N.)

**Keywords:** tumor heterogeneity, patient-derived organoids, circulating tumor cells, breast cancer, treatment resistance, tumor microenvironment, epithelial–mesenchymal transition

## Abstract

Breast cancer is a highly heterogeneous disease, with tumors capable of adapting to shifting conditions, making the development of effective personalized therapies particularly challenging. Patient-derived models, such as patient-derived organoids (PDOs) and circulating tumor cell (CTC) cultures, have emerged as powerful tools for investigating intra- and inter-tumor heterogeneity. These models largely retain the genetic, phenotypic, and microenvironmental features of the original tumors, providing valuable insights into disease progression, drug response, and resistance mechanisms. Furthermore, by enabling tumors’ spatiotemporal molecular profiling, PDOs and CTCs offer a dynamic approach to assess treatment efficacy over time. However, to fully capture the complexity of breast cancer heterogeneity, it is required to develop models from multiple tumor and blood samples collected throughout the course of treatment. This review explores the potential of integrating PDOs and CTC models to better understand intra-tumor heterogeneity while addressing key challenges in developing patient-derived models that accurately recapitulate patients’ tumors to advance personalized care. The integration of PDOs and CTCs could represent a paradigm shift in the personalized management of metastatic breast cancer.

## 1. Introduction

Breast cancer remains the most frequently diagnosed cancer and the second leading cause of cancer-related deaths among women [1]. While advances in early-stage adjuvant therapies and targeted treatments for metastatic breast cancer have significantly improved patient survival, the development of treatment resistance remains a significant challenge, leading to incurable disease in the metastatic setting [2,3]. Breast cancer classification into clinical subtypes is crucial for guiding treatment selection, but high molecular diversity within each breast cancer subtype, defined as inter-tumor heterogeneity, significantly contributes to treatment resistance [4,5]. Furthermore, there is significant variability in cancer cells within a single tumor or patient, known as intra-tumor heterogeneity [6,7]. In metastatic disease, more aggressive cell populations can emerge in different metastatic sites, adapt to distinct microenvironments, and spread to new locations [8,9,10]. Targeted therapies can exacerbate this heterogeneity by selecting resistant cell clones [11]. Furthermore, immune-related events within the tumor microenvironment (TME), such as the diversity of tumor-infiltrating lymphocytes (TILs) and the expression of programmed cell death protein-1 (PD-1) and ligand-1 (PD-L1), can vary across the different metastasized organs within a patient. This variability may contribute to distinct responses to immune checkpoint inhibitors in different metastatic sites [12].

Currently, tumor histopathological analysis and molecular testing often rely on a single biopsy, which fails to capture the full genetic diversity within solid tumors, as tumors exhibit significant genetic heterogeneity across different regions and metastatic sites. Additionally, the evolving nature of tumors over time makes a single biopsy insufficient to represent the dynamic genetic landscape (Figure 1). Therefore, obtaining samples from multiple metastatic sites and longitudinal blood or tumor samples is crucial for comprehensively analyzing the tumor’s genetic profile and selecting more effective, personalized treatment strategies.

Patient-derived organoids (PDOs) are three-dimensional (3D) cellular structures that can self-renew and differentiate and, when developed from different regions of the primary tumor and multiple metastatic sites, enable studying single-cell heterogeneity and functional diversity within a single patient. Moreover, circulating tumor cells (CTCs) obtained from liquid biopsies offer a non-invasive approach to monitor real-time treatment responses and assess tumors’ genomic, transcriptomic, and proteomic diversities [13,14,15]. This review explores the opportunities and challenges of the complementary use of matched patient-specific PDOs and CTCs to study tumor heterogeneity and treatment resistance in advanced breast cancer.

## 2. Tumor Heterogeneity in Metastatic Breast Cancer

Breast cancer is a highly heterogeneous disease characterized by diverse histopathological and molecular subtypes that can evolve, both as the disease progresses and after treatment-induced selective pressure. The dynamic spatial and temporal diversity within an individual patient’s tumor (intra-tumor heterogeneity) and across different patients with the same cancer subtype (inter-tumor heterogeneity) contribute to distinct treatment responses and the development of treatment resistance, either intrinsic (preexisting resistance mechanisms) or acquired (resistance mechanisms emerging during treatment) [16,17]. Thus, developing successful personalized therapies requires a deeper understanding of tumor heterogeneity and access to preclinical models that recapitulate this complexity.

Breast cancer is traditionally classified by immunohistochemistry into three clinical subtypes: hormone receptor-positive (HR+)/human epidermal growth factor receptor 2 negative (HER2−), HER2+, and triple-negative breast cancer (TNBC). This classification remains the one in use to guide current therapeutic strategies for breast cancer, which provides only a broad measure of inter-tumor heterogeneity and results in insufficiently tailored treatments [4]. Molecular profiling has further refined breast cancer subtypes into four main groups, luminal HR+ A and B, HER2-enriched, and basal-like, while also identifying distinct transcriptomic subgroups within TNBC [18]. Indeed, more than 20 distinct subtypes that differ genetically, morphologically, and clinically, and over 1600 driver mutations have been identified in breast cancer, highlighting its heterogeneous nature [19,20]. Nevertheless, bulk tissue molecular profiling used in these studies has limitations in assessing intra-tumor heterogeneity [12]. Single-cell analysis is needed to understand tumor genomic and transcriptomic instability, which drive the selection and outgrowth of diverse cancer clones, leading to various cellular phenotypes and treatment resistance mechanisms [21,22].

Tumor heterogeneity is also influenced by diverse non-tumor components constituting the TME, such as immune cells, fibroblasts, endothelial cells, adipose tissue, soluble signaling molecules, and extracellular matrix ECM [23]. These interact with cancer cells by secreting and expressing factors that promote tumor growth by suppressing anti-tumor immune response, or block tumor growth by activating anti-tumor immunity [24,25]. The TME varies across breast cancer subtypes in both the quantity and composition of immune cell populations [26]. TNBC has been considered the most “immune-hot” subtype, but HER2-amplified and high-grade HR+ luminal B tumors may also exhibit significant numbers of TILs and increased immune checkpoint expression. In contrast, luminal A tumors have shown to be largely immune-cold [27,28]. An unmet need in cancer research remains the effective capture of tumor and TME heterogeneity in preclinical models that can be functionally tested in vitro. Standardized and reproducible models that reflect intra- and inter-tumor heterogeneity are critical for advancing precision oncology and developing effective personalized therapies.

## 3. Capturing Tumor Heterogeneity with In Vitro Patient-Derived Models

### 3.1. Overview of PDOs in Cancer Research

PDOs are a self-organized, 3D assembly of cancer cells derived directly from a patient’s tumor that have emerged as a promising and cost-effective tool for modeling cancer heterogeneity. PDOs can be composed of tumor cells alone or incorporate the TME, such as immune and stromal cells, either in a native or reconstituted model [29]. PDOs can recapitulate many of the structural, molecular, and cellular features of the original tumor, including tumor subtype markers, mutations, copy number alterations, and cellular architecture. Because PDOs maintain, at least initially, the heterogeneity present in the original tumors, they serve as an excellent tool for studying both inter-tumor and intra-tumor heterogeneity [30]. These models represent a substantial improvement compared to other and more used human-derived breast cancer models, including cancer cell lines and patient-derived xenografts (PDXs), whose development is inefficient, labor-intensive, and time-consuming, making their widespread use for preclinical testing impractical. Furthermore, cell lines and PDXs are typically generated from advanced-stage tumors, limiting their ability to fully represent the cancer spectrum [31]. In breast cancer, PDOs have been shown to preserve the histological and molecular characteristics of the original tumors, including key markers such as estrogen receptor (ER), progesterone receptor (PR), and HER2, copy number variations, and sequence alterations [29]. Notably, breast cancer organoids have been successfully developed across all major gene expression-based subtypes and enable in vitro drug screening, with results that align with in vivo xenotransplantation findings and patient responses [29]. An overview of PDOs used as preclinical models for breast cancer is shown in Table 1.

#### 3.1.1. PDO Culture Methods

Significant progress has been made in recent years in the methodologies for deriving cancer organoids. Advances in tissue sourcing now allow organoid generation from a range of specimens, including surgical resections, core needle biopsies, liquid effusions, and even blood samples. While it is possible to establish PDOs from tumor cells isolated from blood and effusion samples, this review focuses on PDOs derived from solid tumors, which remain the most widely used source. The most commonly utilized cancer organoid model is the submerged culture system, wherein mechanically and enzymatically dissociated tumor cells are embedded in an ECM and cultured under a medium enriched with growth factors and pathway regulators tailored to the specific tumor type [38,39,40]. A key modification in breast cancer organoid culture protocols has been the addition of Neuregulin 1, a ligand for HER3 and HER4 involved in mammary development and tumorigenesis, which enables efficient organoid generation and long-term expansion [31]. The addition of a Rho-associated protein kinase (ROCK) inhibitor further enhanced culture conditions by promoting the survival and proliferation of tumor epithelial cells. Media optimization tests further revealed that Wnt-3A had minimal impact, epidermal growth factor (EGF) concentrations above 5 ng/mL increased proliferation but disrupted organoid integrity, and addition of a p38 MAPK inhibitor reduced organoid establishment efficiency at concentrations above 1 μM [31].

While the submerged culture method enables efficient organoid generation and easy scalability, it is limited in its ability to maintain non-epithelial cells from the TME. To address this limitation, alternative methods have been developed, including the air–liquid interface (ALI) culture system, where minced tumor fragments are directly embedded in collagen within a transwell, with the upper layer exposed to air while the lower layer is in contact with culture media [40,41]. Another method involves first culturing minced tumor tissue in low-attachment plates to form spheroids, which are then transferred to a microfluidic device with controlled flow and regulated concentrations of gel and media through microchannels to provide a stable and dynamic environment for the spheroids [42,43]. Both ALI and microfluidic systems allow studying immunotherapy responses and characterizing immune cell populations within the TME [44,45,46]. The most advanced approach constitutes the organ-on-a-chip system, which integrates microfluidics, tissue engineering, and cell biology to simulate organ-level human physiology [47]. This approach allows precise control of nutrient and signaling gradients and can model complex tissue environments by incorporating multiple cell types, but its high cost, complexity, and need for specialized training and equipment limit widespread use.

#### 3.1.2. Co-Culture Models of Tumor and Tumor Microenvironment

A significant advancement of PDO technology is its ability to model the TME in a patient-specific manner, accurately replicating both the tumor architecture and immune and non-immune composition. This is crucial for testing the growing number of therapies targeting the TME. The TME is a complex and dynamic network of immune, stromal, and endothelial cells, adipose tissue, ECM components, and soluble factors that regulate tumor progression, metastasis, and response to therapy [48,49]. Including immune and stromal cells into PDOs through a reconstituted or native model enables a comprehensive study of TME–tumor interactions and their impact on tumor behavior and treatment outcomes.

The conventional submerged organoid models are easier to generate from cancer tissue, but their lack of stromal components halts the study of TME-targeted therapies. More complex organoid models that incorporate both tumor and non-tumor components have been developed, either by reconstitution of isolated epithelial, stromal, and immune cells in already developed tumor PDOs (bottom-up reconstitution) or by maintaining the native architecture of the tumor by simply mincing the tumor tissue without enzymatic digestion (top-down reconstitution, Figure 2).

Reconstituted TME organoid models are developed by first generating single-cell suspensions from enzymatically and mechanically digested tumor tissue. These tumor cells are initially grown in an ECM and then co-cultured with engineered and/or previously expanded immune cells, such as TILs or autologous peripheral blood immune cells used to expand tumor-reactive T cell clones [50,51]. Monocyte-derived macrophages, have also been successfully co-cultured with human intestinal organoids in chip-based models of inflammatory bowel disease [52]. Tumor organoids and immune cell co-cultures are also promising models for testing the killing capacity of engineered immunotherapies, such as chimeric antigen receptor (CAR) T cells, in solid tumors [53,54]. In breast cancer, 3D organotypic co-culture systems combining natural killer (NK) cells with tumor organoids have been developed to model tumor–immune cell interactions and to test NK cell-mediated cytotoxicity against metastatic breast cancer cells [37]. Another study using organ-on-a-chip technology developed a tri-culture system, including a dual-compartment chip to investigate tumor matrix remodeling, a 3D tubular chip to study bone metastasis using bone marrow-derived stem cells and endothelial cells, and a liver–tumor co-culture chip that mimics hepatic metabolism for drug screening [55]. In contrast to this reconstituted model that establishes tumor PDOs before adding TME components, co-cultures can also be derived directly after enzymatic tumor digestion with non-purified tumor cell suspensions for short-term co-culture of tumor and non-tumor cells [56,57]. While this model retains the patient TME cell populations, dissociation into single cells disrupts the tumor and TME architecture and spatial conformation and may alter crucial cell interactions between tumor and non-tumor cells that affect tumor progression and treatment response [58,59]. Indeed, the new spatial distribution following dissociation into single-cell suspensions may completely remove some tumor–TME functional interactions and introduce new interactions between cell types that were previously separated, as well as expose tumor and non-tumor cells to non-physiological ECM, which can change cancer and non-cancer cell morphology and function [29].

Other approaches, such as the ALI model and microfluidic devices, are used to study the native tumor and TME within their original architecture. In contrast to reconstituted models, these organoid systems allows the growth of tumor cells en bloc alongside endogenous non-tumor cells, retaining both the heterogeneity of cell populations, such as cancer-associated fibroblasts (CAFs), cytotoxic T lymphocytes (CTLs), T-helper cells, tumor-associated macrophages (TAMs), NK cells, and B cells and the tumor architecture and spatial arrangement of its components [42,43,60].

The accurate representation of the 3D tumor architecture, heterogeneity, and treatment response by PDOs containing TME presents an excellent opportunity to test patient-specific responses to immunotherapy. PDOs reconstituted with peripheral blood lymphocytes or TILs have been used to investigate patients’ responses to checkpoint blockade or adoptive cell therapy with variable success [53,61,62,63,64]. Another study explored the role of CAFs in influencing pancreatic ductal adenocarcinoma (PDAC) organoid behavior. This study identified two distinct CAF subtypes, myofibroblastic and immunoinflammatory, that interacted differently with the organoids, highlighting CAF subtype heterogeneity and its impact on tumor progression and therapy response [65]. Notably, CAF-secreted interleukin-6 promotes epithelial–mesenchymal transition (EMT) and drug resistance in esophageal adenocarcinoma, suggesting these as potential biomarkers for improved patient stratification [66]. In breast cancer, a whole-tumor cell (WTC) ex vivo model was successfully established from 98 out of 116 tumor samples, effectively recapitulating the parental tumor architecture and endogenous microenvironment. This model demonstrated strong clinical correlations and predictive value across a wide range of breast cancer therapies [35].

Noteworthily, the utility of TME organoid models for testing immunotherapy response and resistance alongside clinical treatments is limited by their short viability, potentially missing treatment resistance driven by rare tumor clones. Comparing patient-matched PDOs from pre- and post-treatment samples and developing immune organoid models from both primary and metastatic tumors could improve the study of immunotherapy efficacy in late-stage cancers. Furthermore, reconstituted models that incorporate only a single immune subtype may require greater complexity by the addition of several key TME components to improve both short- and long-term predictive accuracy. Lastly, advancements in 3D culture techniques, such as ECM scaffolds, microfluidics, and media optimization, could further optimize and standardize TME organoid models [67].

#### 3.1.3. Applications of PDOs in Drug Testing and Response Prediction

PDOs are a promising tool for drug screening and predicting therapeutic responses. Studies have shown that drug testing in PDOs generally aligns with patient responses, though most investigations have been limited to small cohorts [30]. Although more extensive studies remain scarce, a study with a pancreatic cancer PDO biobank showed an 89% match between PDOs and patient responses [68]. Another study with gastrointestinal tumor organoids successfully predicted outcomes in 20 of 21 cases [69]. Specifically in breast cancer, 12 PDOs were established from needle biopsies of metastatic lesions, and two models, tamoxifen-sensitive and -resistant, were selected for in vitro and in vivo testing. The observed drug responses closely mirrored those of the corresponding patients, demonstrating clinical relevance [31]. Similarly, another study generated PDOs from eight breast cancer patients, including diverse molecular subtypes and cases of multidrug resistance. These PDOs demonstrated drug response profiles that were consistent with patient clinical outcomes, revealing either sensitivity or resistance to a wide range of therapies, including epirubicin, paclitaxel, docetaxel, cyclophosphamide, gemcitabine, vinorelbine, trastuzumab, pyrotinib, olaparib, 5-fluorouridine, etoposide, and alpelisib [32]. These findings highlight the potential of PDOs to inform and guide personalized treatment decisions. Furthermore, PDOs are useful for drug resistance mechanisms studies by evaluating resistant clones, which may harbor mutations or altered signaling pathways that help them evade therapy. Identifying these pathways can inform the development of combination therapies, where one drug targets a key pathway central for the resistant clones while another targets central pathways for both resistant and sensitive cells. PDOs also provide an excellent platform for studying tumor metastasis. By exposing PDOs to different treatment regimens and monitoring their growth and survival, researchers can better understand the dynamics of metastatic spread and the role of intra-tumor heterogeneity in facilitating metastasis. These studies help elucidate how specific clones within a tumor are better equipped to survive in different microenvironments, which is crucial for developing more effective treatments that target metastatic disease.

However, organoids can change in culture, as subclonal mutations may be gained or lost over passages, while truncal mutations generally remain stable [70,71]. Indeed, early-passage organoids closely reflect the mutational profile of the original tumor tissue, but genetic drift and treatment selection pressure in culture often lead to divergence over time [29]. Moreover, gene expression and epigenetic alterations may be influenced by culture conditions, which do not fully replicate the in vivo environment [29]. Furthermore, organoids represent only the tumor sample from which they were derived, limiting their ability to capture the full spectrum of the tumor heterogeneity. A more comprehensive approach requires culturing samples from multiple tumor regions and distinct metastatic sites.

Importantly, organoids derived from different metastatic sites of a patient with pancreatic cancer showed comparable responses to some chemotherapies but varied significantly with others [68]. Similarly, PDOs developed from different regions of the same colorectal tumor exhibited up to a 30-fold variation in treatment responses, underscoring the need for multiple sampling to capture intra-tumor heterogeneity fully [72]. Notably, a study analyzing organoids from 27 samples across five liver cancer patients found relatively low variability in drug response between patients but significant differences among samples from the same patient [29]. Additionally, a large colorectal cancer study involving multi-omic analysis of 78 clonally derived organoids from multiple tumor regions of three patients revealed substantial variability in mutations, gene expression, epigenetic modifications, and drug responses among clonal organoids from the same region [73]. Together, these studies highlight the potential of PDOs in capturing key aspects of tumor heterogeneity and predicting treatment responses but underscore the need for sampling multiple tumor regions and metastatic sites to represent intra-tumor heterogeneity accurately. Furthermore, the lack of standardization in organoid development/culture methodologies remains a major source of variability, impairing reproducibility and comparability across studies and limiting their clinical application [29]. Implementing PDOs in real-world clinical settings faces additional challenges, including the time-consuming process of establishing and testing organoids that may delay treatment decisions, the high costs associated with specialized laboratory infrastructure and skilled personnel, and logistical difficulties in promptly acquiring and processing fresh tumor samples. Together, these limitations hinder the broad clinical use of PDO-based approaches.

### 3.2. Overview of CTCs in Metastatic Cancer

CTCs are cancer cells shed from the primary tumor or metastasis into the bloodstream and play a key role in the metastatic process by intravasation and extravasation from the circulation and colonization of distant organs [74,75] (Figure 3). CTCs are extremely rare, with numbers in peripheral blood of about 1 CTC per 1–10 million white blood cells [76]. In contrast to tumor resections and biopsies, CTCs can be collected non-invasively and repeatedly, enabling longitudinal monitoring of spatial and temporal changes in the tumor’s genetic and phenotypic profiles throughout disease progression and treatment response [77]. Thus, CTCs hold exceptional potential in prognosis prediction, real-time monitoring of treatment response, and personalized drug susceptibility testing [78].

CTCs exhibit high heterogeneity and a wide range of molecular alterations, including mutations, gene expression alterations, and epigenetic modifications, contributing to their ability to survive in circulation and colonize distant organs. Despite their heterogeneity, all metastasis-competent CTCs can self-replicate similarly to stem cells [79]. To maintain a balance between migration and proliferation, metastasis-competent CTCs do not become completely mesenchymal but acquire sufficient mesenchymal features, defined as a partial or hybrid EMT state [80]. This plastic and transient state enables cells to co-express both epithelial and mesenchymal markers, enhancing their tumorigenicity, stemness, metastatic potential, and therapy resistance [81].

#### 3.2.1. CTCs Isolation

In contrast to other liquid biopsy components, such as circulating tumor DNA (ctDNA) and tumor-derived extracellular vesicles (tdEVs), CTCs are a “complete” biological material that can be used to isolate DNA, RNA, and protein simultaneously. Furthermore, in vitro culture of isolated CTCs, although challenging, enables functional studies on metastasis development and drug testing. However, the isolation of CTCs poses a significant challenge due to their rarity and heterogeneity. Most separation techniques rely on either the physical properties, such as size and deformability, or biological characteristics, such as surface markers, of CTCs (Figure 4). Advances in microfluidics, size-based filtration, and immunomagnetic capture have improved the efficiency and reliability of CTC isolation from patient blood samples, but no current method can consistently isolate a pure population of CTCs [74,82,83]. Limitations such as low recovery rates, poor purity, and reduced cell viability, continue to restrict the expansion of CTCs in vitro or in vivo, limiting their use in downstream functional analyses [84].

CTC isolation methods based on physical properties take advantage of distinct characteristics of these cells, including their larger size, greater stiffness, unique density, and dielectric properties compared to other blood cells [75,85]. Size-based enrichment approaches, including filtration methods, separate CTCs, typically 20–30 μm in diameter, from smaller blood cells (8–12 μm). Filtration-based approaches are simple, cost-effective, and high-throughput, but due to heterogeneity of the size and deformability among CTCs, smaller or more flexible tumor cells may be lost, making it difficult to define optimal thresholds. Common size-based systems include ISET (Isolation by Size of Epithelial Tumor Cells), ScreenCell, and the FDA-cleared Parsortix system, which combines microfluidics and stepped physical structures to enhance capture efficiency [86,87,88]. Other physical-based methods use density differences between CTCs and blood cells for separation by density gradient centrifugation with media such as Lymphoprep or Ficoll-Paque. This method enables simple and rapid enrichment of label-free, viable CTCs. However, due to overlapping densities between CTCs and mononuclear cells, the yield of highly pure CTCs is low, often requiring additional purification steps [89]. Moreover, CTCs exhibit distinct dielectric properties compared to other peripheral blood cells, and thus, when exposed to alternating electric fields, their polarization behavior differs, enabling spatial separation from other cells [90]. An example of this approach is dielectric electrophoresis field-flow fractionation (depFFF), which enables label-free CTC isolation based on dielectric differences [91].

Bioaffinity-based separation methods utilize immunoreactions between antibodies, aptamers, peptides, or small molecules and target proteins or receptors on the surface of the CTCs to enable their recognition, enrichment, and isolation [90]. Positive selection with antibodies against EpCAM or cytokeratins, or negative selection of leucocytes with anti-cluster of differentiation CD45 antibody, followed by separation of antibody-tagged cells with magnetic beads, are commonly used to separate viable CTCs [75]. The FDA-approved assay CellSearch identifies CTCs based on cytokeratin expression, lack of CD45, and nuclear labeling with DAPI [75]. Furthermore, this method enables the collection of CTC-clusters, tdEVs, and apoptotic bodies, and the antibodies used can be replaced by antibodies targeting other antigens of interest. Similarly, the CellCollector allows the detection and isolation ex vivo of EpCAM-expressing cells [92]. Noteworthy, selection based on EpCAM and cytokeratin levels may only capture a subset of CTCs, as these may exhibit a highly variable expression of epithelial markers. Indeed, CTCs with mesenchymal or basal-like phenotypes may exhibit extremely low levels of epithelial markers [93,94]. Furthermore, cells processed with CellSearch must be permeabilized before staining with antibodies, limiting the downstream applications of the isolated CTCs. As a result, antigen-agnostic methods such as Parsortix remain valuable, as these allow the separation of CTCs based on size and mechanical properties, such as deformability, hereby facilitating downstream expansion both in vitro and in vivo [95,96,97]. Notably, xenografting of CTCs directly after isolation or following expansion in vitro has also been performed to further expand the CTC numbers [98,99]. Breast cancer CTCs have been subcutaneously injected into the mammary fat pad of SCID mice and successfully formed tumors expressing hormone receptors that matched the patients’ tumors [100,101]. Interestingly, it has been shown that breast cancer CTC lines expanded in vivo were capable of metastasizing at the same sites as in the patients the CTCs were isolated from [102]. Currently, 37 breast cancer cell lines derived from CTCs have been reported, of which 11 cell lines were cultured long-term for over 6 months [100,102,103,104,105,106,107].

In addition to EpCAM and cytokeratin, various unique surface proteins expressed on CTCs can be targeted to distinguish them from other cells. By exploiting the specific binding interactions between these proteins and their ligands or substrates, CTCs can be selectively captured and subsequently released [90]. An example is the folate receptor (FR), a cysteine-rich cell surface glycoprotein that binds folic acid (FA) with high affinity, which is minimally expressed in most normal tissues but overexpressed in many cancers to support rapid cell division under low folate conditions. This makes FR a useful target for CTC enrichment and isolation, and has been used to develop a β-cyclodextrin-functionalized poly(acrylamide) hydrogel (GelPAM-CD) with FA conjugated to adamantane-PEG, forming the final scaffold GelPAM-CD/PEGFA for selective CTC capture [108]. Similarly, the small-molecule peptide NGR (asparagine-glycine-arginine) targets aminopeptidase N (CD13), a tumor cell membrane protein, enabling CTC capture in blood [109]. Additionally, aptamers, short DNA or RNA sequences that fold into specific structures, offer another approach to target molecules on the CTC surface. Compared to antibodies, aptamers have advantages including high affinity, small size, low cost, ease of synthesis, and precise target recognition. Furthermore, unlike antibodies, aptamers enable gentle CTC release by temperature or hybridization-induced conformational changes, preserving cell integrity post-capture. These properties have been used to design devices such as the fluidic multivalent grafted nanointerface microfluidic chip (FLASH-Chip) and the aptamer-functionalized octopus chip (AP-Octopus-Chip) [110,111]. While physical and bioaffinity-based methods capture CTCs, detection rates vary due to differing identification criteria. Indeed, the FDA-approved CellSearch system correlates CTC count with prognosis but is limited clinically because its reliance on EpCAM+, CK+, DAPI+, CD45− markers may miss CTCs undergoing EMT, which often show reduced EpCAM and CK expression. To overcome these limitations, integrated microfluidic chips combining multiple detection methods are being developed to enable high-efficiency capture, intact protein profiling, compatibility with single-cell sequencing, and live-cell drug testing. Additionally, because CTC counts in small blood samples are low, in vivo enrichment technologies are emerging to sample broader circulatory volumes, enhancing CTC yield for downstream analyses.

#### 3.2.2. CTCs Prognostic and Predictive Potential in Breast Cancer

The FDA-approved CellSearch system has supported CTCs as a strong and independent prognostic biomarker for early and metastatic breast cancer [112]. In the neoadjuvant setting, the prognostic significance of CTCs detected by CellSearch was evaluated in the GeparQuattro trial in patients with operable or locally advanced HER2+ breast cancer, both before and after neoadjuvant therapy with HER2-targeted therapy together with anthracycline-taxane-based chemotherapy. Shorter disease-free survival (DFS) and overall survival (OS) were observed for ≥1 or ≥2 CTCs/7.5 mL before neoadjuvant chemotherapy, while no significant association was found with the CTC numbers after neoadjuvant chemotherapy [113]. In the adjuvant setting, the presence of CTCs was assessed with CellSearch before and two years after chemotherapy in early-stage, high-risk breast cancer patients enrolled in the phase III SUCCESS A trial [114]. This trial compared two adjuvant chemotherapy regimens followed by either two or five years of zoledronate. Detection of CTCs two years post-chemotherapy was independently associated with poor OS and DFS, regardless of CTC status at baseline. Notably, patients who were CTC-positive both at baseline and at the two-year follow-up showed the poorest survival outcomes [114]. However, undetectable levels of CTCs for most early-stage breast cancer patients hinders the clinical utility of CTCs in this setting, and for those with detectable levels, the low numbers complicate molecular testing [115]. Nevertheless, baseline CTC detection remains a strong predictor of poor prognosis, supporting treatment escalation in high-risk groups. Noteworthily, since not all CTCs possess metastatic potential, and given the fact that those that do may not survive in the bloodstream, CTC detection does not necessarily indicate poor clinical outcomes [116]. In the metastatic setting, a prospective multicenter study evaluated 177 patients with measurable metastatic breast cancer for CTC levels using CellSearch, both before starting a new line of treatment (baseline) and at the first follow-up visit. CTC levels at both time points independently predicted progression-free survival (PFS) and OS, independently of the time to metastasis, site of metastasis, or HRstatus. Importantly, CTC levels predicted poor OS and PFS earlier than traditional imaging methods, with changes detectable three to four weeks after therapy initiation compared to eight to 12 weeks by imaging [82]. Additionally, CTC dynamics after treatment can inform prognosis, as patients with high baseline CTC counts but rapidly decreasing numbers during treatment have a favorable prognosis [117]. However, despite their prognostic potential, CTCs are not yet a validated standard for guiding therapeutic decisions due to insufficient standardization.

Beyond their prognostic value, CTC enumeration and surface marker expression have also been explored for their predictive potential [118]. The STIC CTC study, a multicenter randomized phase III trial, compared first-line therapy (ET or chemotherapy) based on CTC count (CTC arm) versus the investigator’s choice (standard arm). The trial showed that the CTC-guided approach was noninferior to clinician-driven treatment in terms of 2-year PFS. While no OS benefit was observed in the general population, patients with a high CTC count but low clinical risk experienced a significant OS advantage from chemotherapy, highlighting the potential of CTC counts to guide the choice between ET and chemotherapy [118,119,120]. 

Notably, several studies have investigated whether HER2+ CTCs in HER2- metastatic breast cancer could predict response to HER2-targeted therapy. A multicenter phase II trial evaluated the efficacy of lapatinib in metastatic breast cancer patients whose primary tumors were HER2- but had HER2+ CTCs, as isolated by CellSearch and assessed for HER2 status by immunofluorescence (IF) [121]. CTC positivity was defined as ≥2 CTC per 7.5 mL of blood, and HER2 positivity as ≥50% of the CTCs expressing HER2. Among the 139 HER2- patients screened, 96 patients were CTC+ (median number of CTCs: 19; range 2–1637) and seven of these (7%) had HER2+ CTCs and were eligible for treatment with lapatinib. No objective tumor responses were observed although one patient achieved disease stabilization for 8.5 months. Similarly, the DETECT III trial investigated the benefit of HER2-targeted therapy in HER2- metastatic breast cancer patients with HER2+ CTCs [122]. Of 2137 patients screened, 101 were eligible and randomized to receive standard therapy with or without lapatinib. The primary endpoint was CTC clearance (absence of CTCs at the end of treatment), while secondary endpoints included PFS, OS, and safety. Although CTC clearance rates did not significantly differ between groups, patients treated with lapatinib had an improved OS compared to the standard therapy group (20.5 vs. 9.1 months, *p* = 0.009).

The CirCe01 trial assessed the clinical utility of CTC-based monitoring in metastatic breast cancer patients beyond the third-line of chemotherapy. Patients with ≥5 CTC per 7.5 mL, as determined by CellSearch, were randomized to standard arm or CTC-arm, where CTC counts were evaluated after each chemotherapy cycle, and early tumor progression, predicted by rising CTCs, led to switching to subsequent therapies. The trial found no difference in OS between the two arms and failed to demonstrate the clinical utility of CTC monitoring [121]. Overall, trials evaluating the predictive value of CTCs have shown inconsistent results, and further and more robust studies are needed to establish their definitive clinical role [120,121,122,123]. An overview of these studies is shown in Table 2.

#### 3.2.3. CTCs Molecular and Functional Analysis

Once isolated, CTCs can undergo various analyses beyond enumeration, including molecular profiling techniques, such as single-cell RNA and DNA sequencing, to gain insights into their genetic and phenotypic profiles. Whole-genome sequencing of CTCs can identify mutations shared with the primary tumor, as well as novel subclonal mutations that may drive metastasis and treatment resistance. However, due to the low number of CTCs, cell expansion is necessary for functional studies. These studies may help to assess the most aggressive, metastatic cells, those with self-renewal capacity in culture, while also evaluating their molecular profiles [92]. Additionally, the ex vivo culture of CTCs may be used to assess the efficacy of specific treatments and investigate the development of drug resistance mechanisms [124]. Advances in microfluid chips have allowed the quick process of whole blood samples for simultaneous CTC isolation and drug testing [125]. Indeed, CTC cultures derived from patients with metastatic ER+ breast cancer were tested by next-generation sequencing and with a panel of targeted inhibitors, including PI3K, FGF, IGF, and ER targeting drugs, as well as cytotoxic agents, including doxorubicin, paclitaxel, and capecitabine [100]. Notably, the CTC line exhibiting a *PIK3CA* mutation showed high sensitivity towards the PI3Kα isoform-specific inhibitor BYL719 in vitro and in vivo [100]. In another study, the cyclin-dependent kinase 4 and 6 (CDK4/6) inhibitor palbociclib was tested in a CTC cell line compared to MCF7 cells, which showed sensitivity of CTCs to this drug, supporting CDK4/6 inhibitor clinical benefit for this patient [102].

Multi-modal single-cell assays can capture the complete CTC profile, including its genome, transcriptome, proteome, and epigenome. Additionally, multiparametric analysis of various liquid biopsy components, such as ctDNA and tdEVs, can provide complementary tumor information, particularly in early-stage disease, where CTC numbers are low. The short lifespan of CTCs in circulation suggests continuous renewal and release, whereas ctDNA primarily reflects dead tumor cells and passive cellular debris shedding. As a result, ctDNA is better suited for detecting minimal residual disease and identifying gene alterations than CTCs [126].

## 4. Integrating Patient-Derived Models from Solid and Liquid Biopsies to Study Tumor Heterogeneity in Metastatic Breast Cancer

### 4.1. Opportunities for Combined Use of PDOs and CTCs

While tumor and liquid biopsies have been extensively molecularly profiled independently, their integration offers a more comprehensive view of tumor heterogeneity in metastatic breast cancer. Integrating functional drug testing using PDOs generated from tumor samples with real-time genomic monitoring of CTCs isolated from liquid biopsies could enable dynamic and personalized treatment strategies. Importantly, concurrent analysis of both solid and liquid biopsies may be required to detect variants missed by either method alone [127]. Studies have shown that over half of the mutations detected in either tissue or liquid biopsy are not common across biopsies, highlighting their complementary potential [128,129]. In fact, the addition of plasma testing to tissue increases therapeutic target detection for patients with metastatic non-small-cell lung cancer by 15% [130]. Moreover, concordance among frequent alterations varies across genes and decreases with longer time intervals between tissue and blood collection [131]. Specifically, *ESR1* variants are more likely to be detected in circulating cell-free DNA (cfDNA) than tissue, whereas certain variants in *PIK3CA* are more commonly identified in tissue samples, suggesting that a combined approach improves detection of alterations [127]. Indeed, while driver events are frequently shared between multiple biopsies, acquired alterations detected in liquid biopsies are frequently polyclonal, present at low allelic fractions, and are indicative of multi-clonal convergent evolution. Furthermore, significant genomic variability has also been observed across primary tumors, relapses, and metastatic biopsies in patients with endocrine therapy resistance [132]. This intra-tumor heterogeneity poses clinical challenges for selecting optimal treatments and identifying resistance mechanisms. Therefore, integrating matched patient-derived models from serial solid and liquid biopsies offers a promising strategy to capture and understand the spatiotemporal dynamics of intra-tumor heterogeneity.

Individually, both PDOs and CTCs offer unique insights into tumor behavior (Table 3). PDOs model the 3D architecture and genetic landscape of the tumor, while CTCs allow for real-time assessment of metastatic potential and the clonal diversity of cancer cells circulating in the bloodstream. Integrating these two models enables a holistic view of tumor heterogeneity and evolution, providing a platform to monitor how breast cancer adapts under therapeutic pressure (Figure 5). Notably, pairing PDOs with CTC analyses from the same patient enables tracking of the tumor’s spatial and temporal clonal evolution, offering insights into the adaptive strategies that cancer cells use to survive treatment. Growing evidence supports the concept of metastatic plasticity in breast cancer, including clinical reports of atypical metastatic patterns, such as duodenal involvement, underscoring the need for patient-derived models capable of capturing both clonal diversity and spatial heterogeneity [133]. The combined use of PDOs and CTCs may provide a more accurate representation of metastatic behavior in real-time, supporting more precise therapeutic decision-making. The emergence of treatment resistance mechanisms in PDOs can be compared to the corresponding CTC profiles to pinpoint which clones demonstrate resistance and invasion abilities and how these clones respond to subsequent treatment cycles. This cross-model tracking allows mapping the clonal selection and adaptation process under therapeutic pressure, providing valuable data on the persistence of resistant cell populations and the likelihood of specific clones contributing to disease progression.

This complementary approach may also help identify new molecular targets for combination therapies by finding specific gene expression patterns or resistance markers shared across PDOs and CTCs. Testing new drug combinations on resistant PDO clones and profiling CTCs for survival markers may contribute to more robust and personalized treatment strategies. In addition, by identifying and targeting resistant clones early, it may be possible to reduce relapse rates and improve long-term outcomes in breast cancer. This approach enhances the predictive power of preclinical models and identifies potential therapeutic targets that might otherwise be overlooked when using PDOs or CTCs separately.

As the field of precision oncology advances, the combined use of PDOs and CTCs can become an integral part of clinical decision-making, particularly in the metastatic setting. This dual-model approach offers a rapid, iterative method to test and refine treatment regimens based on both initial tumor-derived organoid characteristics and real-time responses in CTCs. The adaptability of these models to incorporate new genetic information or emerging resistance mechanisms may enable clinicians to adjust treatments proactively, thus likely improving patient outcomes and reducing resistance-driven progression. Moreover, as larger datasets from PDO and CTC studies become available, these insights can be leveraged to develop predictive algorithms, enabling oncologists to choose the most effective treatment regimen for the individual patient.

### 4.2. Limitations and Future Directions

PDOs and CTCs are undeniably crucial tools for translational cancer research, but significant challenges regarding their use remain. In the case of PDOs, these are difficult to maintain long-term, and their growth may favor specific clones, leading to potential bias in drug testing. Although early passage PDOs often maintain the genetic composition of the original tumor, the extent of clonal selection and genetic drift in later passage PDOs, as well as following freezing-thawing cycles, might be a challenge for some tumors [30,134]. Another major limitation of traditional PDOs is the lack of key tumor-associated components, such as immune cells and fibroblasts, highlighting the need to develop organoid models that more faithfully replicate the TME to advance precision cancer therapy [30]. This variability is further influenced by the complex media composition required for the growth of PDOs. The use of conditioned media containing factors such as Wnt3a, R-spondin, and Noggin and the presence of serum can impede long-term culture viability due to their complexity and cost-effectiveness [135]. Importantly, a key challenge is the lack of standardization of organoid media across different laboratory batches, which may lead to selective pressure favoring specific cell populations and biased results. Recently developed, advanced artificial Wnt agonists may offer promising, cost-effective, standardized, serum-free alternatives to previous Wnt-conditioned media [136,137]. Regarding CTCs, these are rare cells that are difficult to isolate in sufficient quantities for comprehensive analysis and require special culture conditions that are difficult to replicate in vitro. Furthermore, only a small subset of CTCs in the bloodstream may contribute to metastasis formation.

Various studies have shown the potential of PDO and CTC functional drug testing for predicting patient responses to treatment, though more extensive prospective validation is still required. Importantly, these assays should be complemented with clinical measurements of survival outcomes. The establishment of PDO and CTC biobanks representing various cancer subtypes could accelerate drug screening and facilitate the identification of therapeutic response patterns in patients with similar tumor molecular profiles. Furthermore, genome-editing techniques could enable genome-wide functional studies, expand target screening, and enhance small molecule testing beyond traditional drug panels, thereby broadening the range of treatments tested [138,139,140,141]. Scaling up PDOs and CTC cultures in ultra-high-throughput systems would further enhance the feasibility of both genetic and drug testing approaches.

## 5. Conclusions

PDOs and CTCs are revolutionizing the study of breast tumor heterogeneity. These models offer unique insights into genetic, morphological, and microenvironmental diversity that underly cancer progression, metastasis, and treatment resistance. By integrating advanced molecular profiling of PDOs and CTCs, it will be possible to achieve a deeper understanding of the clonal evolution of breast cancer and develop more effective and personalized treatment strategies. Although further clinical validation is needed, PDOs and CTCs represent key tools for the effective implementation of precision oncology in advanced breast cancer.

## Figures and Tables

**Figure 1 jpm-15-00271-f001:**
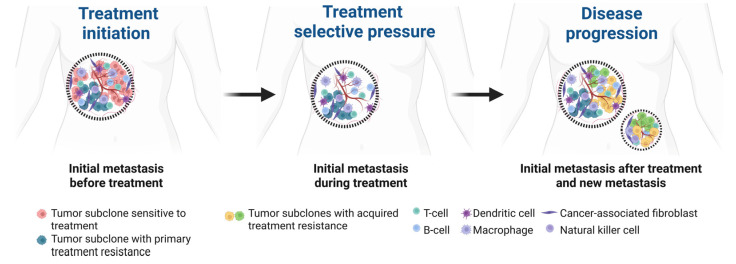
Spatial and temporal heterogeneity of metastatic breast cancer. The initial metastatic site harbors a subclone sensitive to therapy (red) and a second subclone with primary treatment resistance (blue). Under treatment selective pressure, new subclones with acquired resistance and enhanced metastatic potential (green and yellow) emerge within the initial metastatic site and subsequently disseminate to establish secondary metastatic sites. Created in BioRender.com.

**Figure 2 jpm-15-00271-f002:**
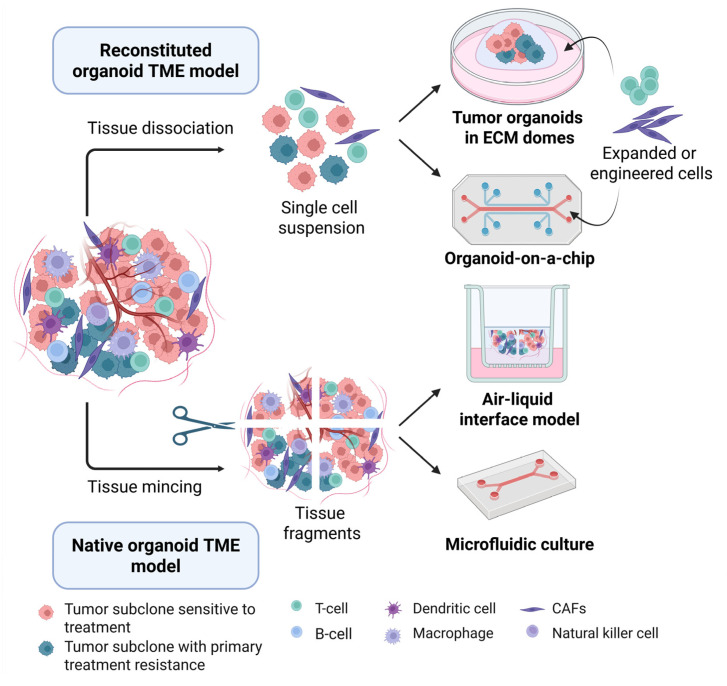
Development of tumor organoid culture systems to model the TME. Reconstituted organoid TME models (**top**), including submerged cultures and organoids-on-a-chip, are derived from tumor single-cell suspensions and subsequently co-cultured with selected cell populations of interest, such as engineered or isolated and expanded immune and stromal cells, to precisely control the composition of the microenvironment. Native tumor TME models (**bottom**) are generated by physically mincing or gently dissociating tumor tissue into fragments, then culturing the fragments in an air-liquid interface or microfluidic culture system, which preserve endogenous immune and stromal cells for a limited period. Abbreviations: CAFs, cancer-associated fibroblasts; ECM, extracellular matrix; and TME, tumor microenvironment. Created in BioRender.com.

**Figure 3 jpm-15-00271-f003:**
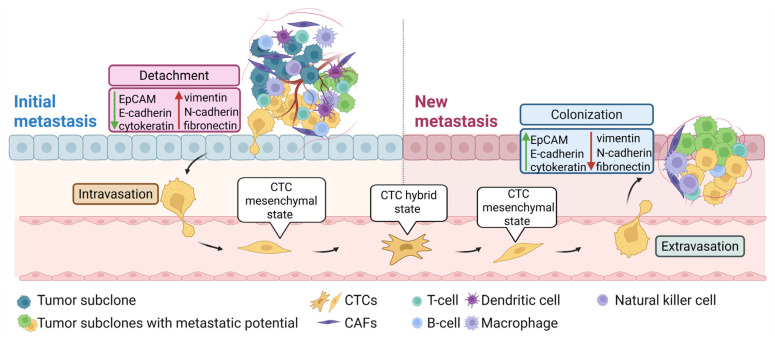
CTCs and EMT in the metastatic process. Tumor cells at the initial metastatic site undergo EMT to invade surrounding tissue, enabling their entry into the bloodstream (intravasation). This process involves upregulation of mesenchymal markers, such as vimentin, N-cadherin, and fibronectin and downregulation of epithelial markers, such as EpCAM, E-cadherin, and cytokeratin. In the bloodstream, CTCs adopt a partial or hybrid epithelial–mesenchymal state, enhancing their metastatic potential. Upon reaching a distant site, mesenchymal CTCs exit the bloodstream (extravasation), invade the new tissue, and revert to an epithelial state to establish a metastatic colony. Abbreviations: CAFs, cancer-associated fibroblasts; CTCs, circulating tumor cells; and EMT, epithelial–mesenchymal transition. Created in BioRender.com.

**Figure 4 jpm-15-00271-f004:**
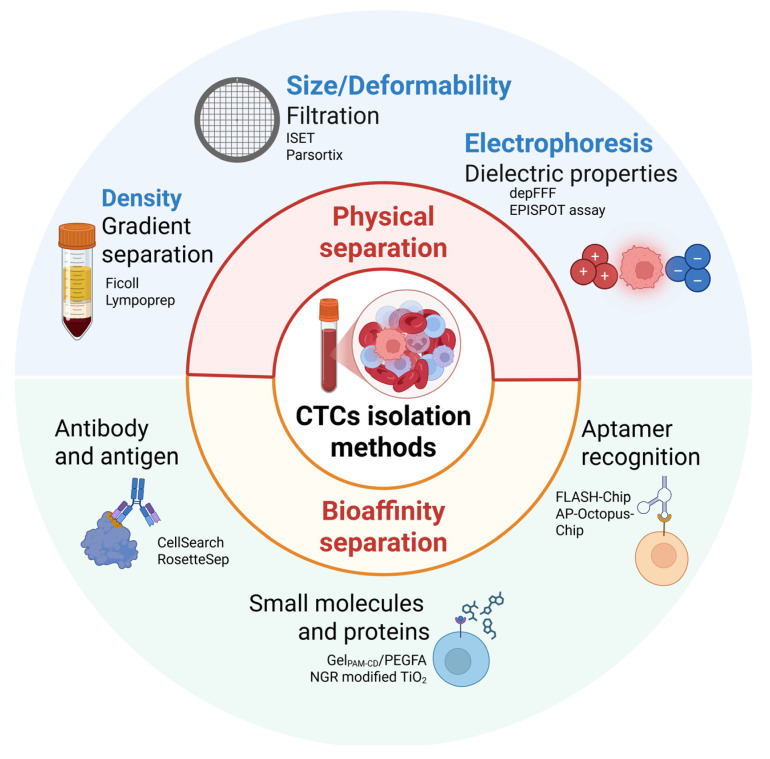
Overview of CTC isolation methods. CTC isolation and enrichment methods are based on either physical properties, such as size, density and dielectric characteristics, or immunoaffinity approaches, which rely on specific binding to antibodies, small molecules, or aptamers. Abbreviations: AP-Octopus-Chip, aptamer-functionalized octopus chip; CTCs, circulating tumor cells; depFFF, dielectric electrophoresis field flow fractionation; FLASH-Chip, fluidic multivalent grafted nanointerface microfluidic chip; Gel_PAM-CD_, β-cyclodextrin (β-CD)-functionalized poly(acrylamide) hydrogel; ISET, isolating by size of epithelial tumor cells; PEGFA, polyethylene glycol folic acid; and NGR, asparagine-glycine-arginine peptide. Created in BioRender.com.

**Figure 5 jpm-15-00271-f005:**
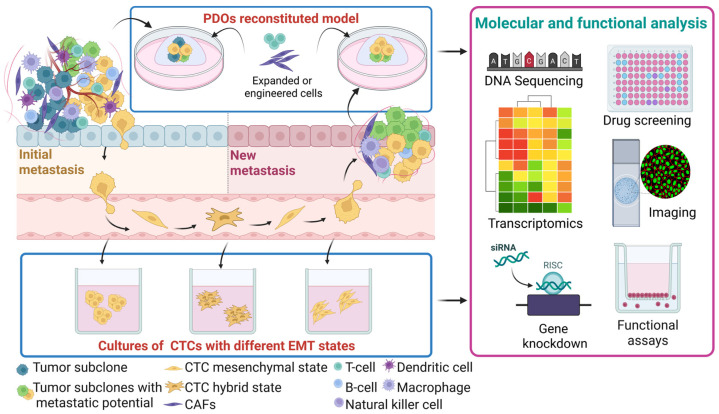
Development of patient-matched PDOs and CTCs to decipher spatiotemporal intra-tumor heterogeneity. PDOs developed from the initial and secondary metastatic sites enable molecular and functional profiling of tumor heterogeneity during disease progression. This approach is complemented by the parallel characterization of matched CTC cultures, providing insights into the most aggressive tumor cell states. Comparative analyses among the initial tumor, disseminated CTCs, and new metastatic lesions provide a dynamic view of tumor evolution. Abbreviations: CAFs, cancer-associated fibroblasts; CTCs, circulating tumor cells; EMT, epithelial–mesenchymal transition; and PDOs, patient-derived organoids. Created in BioRender.com.

**Table 1 jpm-15-00271-t001:** Selected studies on breast cancer PDOs and their applications.

Source Material	Biological Analysis	Applications	Reference
Primary tumor resections and metastatic lesions	RNA-seq, somatic mutation analysis, IF, IHC	Xenotransplantation, drug screening	[31]
Primary tumor resections and metastatic lesions	IHC, WES	Drug screening	[32]
Primary tumor resections	IHC, RNA-seq, WES, WB	Clinical outcome evaluation, drug response prediction	[33]
Primary tumor resections	IHC, WGS	Characterization of organoid-cultured human breast cancer	[34]
Primary tumor resections	WGS, RNA-seq	WTC model of native TME for drug response evaluation	[35]
Locally advanced breast cancer	IHC, IF, WES	Drug testing	[36]
Metastatic lesions	Cytotoxicity study	NK-organoid co-culture, biological research	[37]

Abbreviations: IF, immunofluorescence; IHC, immunohistochemistry; NK, natural killer; RNA-seq, RNA-sequencing; TME, tumor microenvironment; WES, whole exome sequencing; WGS, whole genome sequencing; WTC, whole-tumor cell culture.

**Table 2 jpm-15-00271-t002:** Clinical studies using CTCs detected by CellSearch technology for prognostic or predictive applications in breast cancer.

Application	Stage	Assessment	Outcome	Reference
Prognostic	Operable or locally advanced HER2+ breast cancer	CTC levels before and after neoadjuvant therapy with HER2-targeted therapy plus anthracycline-taxane-based chemotherapy	Shorter DFS and OS for ≥1 or ≥2 CTCs/7.5 mL before neoadjuvant chemotherapy; no significant association after therapy.	[113]
	Early-stage, high-risk breast cancer patients	CTC levels at baseline and after two adjuvant chemotherapy regimens followed by 2 vs. 5 years of zoledronate.	Presence of CTCs 2 years post-chemotherapy was associated with poor OS and DFS, independent of CTC status at baseline.	[114]
	Advanced breast cancer	CTCs levels before starting a new line of treatment (baseline) and at the first follow-up	A CTC level of ≥5 CTC/7.5 mL of blood was an independent predictor of poor OS and PFS on chemotherapy or other systemic therapy.	[82]
Predictive	Early-stage breast cancer	CTC-based vs. investigator’s choice first-line therapy (ET or chemotherapy)	CTC-arm was noninferior to the clinician-arm in 2-year PFS; OS benefit from chemotherapy in patients with high CTCs and low clinical risk.	[118]
	Advanced breast cancer	HER2- primary tumors and HER2+ CTCs treated with ≥1 line of metastatic therapy to receive lapatinib vs. placebo	No objective tumor responses observed; one patient showed disease stabilization for 8.5 months.	[121]
	Advanced breast cancer	HER2- metastatic breast cancer patients with HER2+ CTCs randomized to standard therapy ± lapatinib	No difference in CTC clearance between arms; improved OS with lapatinib compared to standard therapy alone.	[123]
	Advanced breast cancer	Patients with ≥5 CTCs/7.5 mL randomized to standard arm or CTC-guided arm (switch based on CTC counts)	No difference in OS between arms.	[122]

Abbreviations: CTCs, circulating tumor cells; DFS, disease-free survival; ET, endocrine therapy; HER2, human epidermal growth factor receptor; OS, overall survival; and PFS, progression-free survival.

**Table 3 jpm-15-00271-t003:** Overview of the main features of PDOs and CTCs as preclinical cancer models.

Feature	PDOs	CTCs
Source material	Primary tumor biopsies, surgical resections, or metastatic sites (solid biopsy).	Peripheral blood (liquid biopsy).
Frequency of tumor cells in source material	Moderate to high (sample dependent).	Rare (1–10 per 10^7^ white blood cells).
Culture model	3D multicellular structures grown in ECM or synthetic scaffolds.	Single cells or cell clusters cultured in low-attachment conditions (suspension) or 2D adherent cultures. Can also be cultured as 3D CTCDOs.
Culture success rate	Moderate to high (depends on tumor type and technique).	Low to moderate; isolation and expansion remain challenging.
Accessibility	Requires tissue biopsy or surgical sample.	Obtained from blood draw (liquid biopsy).
TME modelling	Reconstituted or native models offer a better representation of the TME, including stromal and immune components.	Do not directly model the TME; primarily reflect disseminated tumor cells. TME may influence CTC release and phenotype.
Tumor heterogeneity	Capture spatial heterogeneity in early passages (site-specific).	Reflect temporal and clonal heterogeneity and metastatic potential of circulating cancer cells, often enriched for highly aggressive or stem-like cells.
Real-time disease monitoring	Limited; requires invasive biopsies.	High; non-invasive and suitable for serial sampling.
Applications	Drug screening, biomarker discovery, precision medicine, tumor heterogeneity, drug resistance studies, modeling tumor evolution, and preclinical drug development.	Real-time disease monitoring, biomarker for early detection of disease or progression prediction, metastasis modeling, drug testing, and real-time tumor genotyping.
Advantages	Allow modeling original tumor architecture, heterogeneity and the TME. High success rate for establishment and can be expanded for high throughput screening and prediction of patient response to therapy.	Minimally invasive. Enables real-time monitoring of tumor evolution or disease progression and captures metastatic potential and drug resistance in the CTCs.
Challenges	Require invasive sampling; difficult standardization; challenging long-term culture (clonal drift or growth inhibition); and time and cost are higher than 2D cultures.	Rare in circulation; low yield; technically difficult isolation and expansion; CTC heterogeneity makes standardized detection/isolation difficult; and the fragility of CTCs affects viability and downstream applications.
Emerging trends	Co-culture with immune cells and stromal components, organ-on-a-chip technologies, and single-cell analysis.	Advanced isolation technologies, single-cell omics, integration with other liquid biopsy components (cfDNA, exosomes), and CTCDOs.

Abbreviations: cfDNA, cell-free DNA; CTCs, circulating tumor cells; CTCDOs, CTC-derived organoids; ECM, extracellular matrix; PDOs, patient-derived organoids; and TME, tumor microenvironment.

## Data Availability

Not applicable.

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
