# Peer review of "Deciphering Breast Tumor Heterogeneity Through Patient-Derived Organoids and Circulating Tumor Cells"

_jpm, 2025, doi:10.3390/jpm15070271_

Round 1
Reviewer 1 Report
Comments and Suggestions for Authors
In this Review, the Authors have critically evaluated the potential of integrating PDOs and CTC models to better understand intratumoral heterogeneity while addressing key challenges in developing patient-derived models that accurately recapitulate patients' tumors to advance personalized care. The work deserves to be published with some minor revisions:
- Can Authors discuss the clinical studies in more detail?
- In Figure 4, please increase the resolution for Transcriptomic heatmap. Text isn’t clear.
- Please improve all figures for clarity.
- More recent references can be added on organoid cultures to highlight the current state.
Author Response
Response to comments by Reviewer #1:
In this Review, the Authors have critically evaluated the potential of integrating PDOs and CTC models to better understand intratumoral heterogeneity while addressing key challenges in developing patient-derived models that accurately recapitulate patients' tumors to advance personalized care. The work deserves to be published with some minor revisions:
1. Can Authors discuss the clinical studies in more detail?
Our response: We appreciate the reviewer’s suggestion to elaborate on the clinical studies. In response, we have expanded Section 3.2.2 CTCs Prognostic and Predictive Potential, where we now provide a more detailed overview of relevant clinical studies (pages 12–15). Additionally, we have included a summary of these studies in the new Table 2 highlighting their design, findings, and relevance to the prognostic or predictive application of CTCs.
2. In Figure 4, please increase the resolution for Transcriptomic heatmap. Text isn’t clear.
Our response: We have improved the resolution of Figure 4 to enhance overall clarity. As the text labels within the transcriptomic heatmap were not relevant and contributed to visual clutter, we have removed them for better readability and visual impact.
3. Please improve all figures for clarity.
Our response: We reviewed all figures in the manuscript to enhance clarity by increasing resolution and enlarging text and labels.
4. More recent references can be added on organoid cultures to highlight the current state.
Our response: We have updated the manuscript to include several recent and relevant references that reflect the current advances in organoid culture methodologies. These additions have been incorporated into the newly organized sections 3.1.1 PDO culture Methods and 3.1.2. Co-culture models of tumor and tumor microenvironment, and Figure 2 (pages 4-7).
Reviewer 2 Report
Comments and Suggestions for Authors
In this Review, Benedetta et al. reviewed patient-derived organoids and circulating tumor cells in breast cancer. They found that PDOs provide tumor static architecture and molecular features, while CTCs dynamically reflect metastatic potential. The combination of the two can comprehensively analyze tumor heterogeneity. Generally, their review work is relatively sufficient. However, some revisions are needed to improve the manuscript.
Here are my comments
- Only four Figures and no Tables in the manuscript. Please add one or two table about PDO and CTC, such as their types, their application, their relationship with tumor microenvironment, publication collection
- There many abbreviations in the manuscript. Please add a section of “Abbreviation List” to facilitate readers’ understanding.
- Pleased add some description of current improvement of PDOs cultivation conditions
- Pleased add some description of CTC separation technology
- In paragraph “4. Integration of PDOs and CTCs to study tumor heterogeneity in metastatic breast cancer”, please reviewed some related publication and add some description: use genomic, transcriptomic, and epigenetic data of PDOs and CTCs with clinical pathological features, construct predictive models, and optimize the selection in personalized medicine.
Author Response
Response to comments by Reviewer #2:
In this Review, Benedetta et al. reviewed patient-derived organoids and circulating tumor cells in breast cancer. They found that PDOs provide tumor static architecture and molecular features, while CTCs dynamically reflect metastatic potential. The combination of the two can comprehensively analyze tumor heterogeneity. Generally, their review work is relatively sufficient. However, some revisions are needed to improve the manuscript.
Here are my comments
1. Only four Figures and no Tables in the manuscript. Please add one or two table about PDO and CTC, such as their types, their application, their relationship with tumor microenvironment, publication collection
Our response: We appreciate the reviewer´s suggestion. In response, we have added a new Figure (Figure 4) which provides an overview of CTC isolation methods. Furthermore, we have included 3 new tables: Table 1 which summarizes key studies of breast cancer PDOs and their applications; Table 2 which detail clinical studies using CTCs for prognostic or predictive applications in breast cancer; and Table 3 which provides a comparative overview of the main features of PDOs and CTCs as preclinical cancer models, such as their culture methods, relevance to tumor heterogeneity, relationship with the tumor microenvironment, and limitations and advantages of each model.
2. There many abbreviations in the manuscript. Please add a section of “Abbreviation List” to facilitate readers’ understanding.
Our response: We have added an Abbreviation List at the end of the manuscript, just before the References (page 19).
3. Pleased add some description of current improvement of PDOs cultivation conditions
Our response: We appreciate the reviewer’s suggestion. In response, we have expanded our discussion of recent advancements in PDO cultivation techniques in a new section 3.1.1. PDO culture methods and section 3.1.2. Co-culture models of tumor and tumor microenvironment, including emerging strategies such as microfluidic platforms and organ-on-a-chip technologies, which enhance physiological relevance and mimic the in vivo tumor microenvironment more accurately (pages 4-7). Additionally, Figure 2 has been updated to reflect some of these methodological improvements.
4. Pleased add some description of CTC separation technology
Our response: We have expanded section 3.2.1. CTCs isolation to include a more detailed description of current CTC separation technologies (pages 9-12). This section now describes both physical- and immunoaffinity-based capture approaches, and includes a brief discussion of their respective strengths and limitations. These methodologies are also illustrated and summarized in the newly added Figure 4.
5. In paragraph “4. Integration of PDOs and CTCs to study tumor heterogeneity in metastatic breast cancer”, please reviewed some related publication and add some description: use genomic, transcriptomic, and epigenetic data of PDOs and CTCs with clinical pathological features, construct predictive models, and optimize the selection in personalized medicine.
Our response: We thank the reviewer for this suggestion. In response, we have revised section 4.1 Opportunities for combined use of PDOs and CTCs to include relevant studies comparing genomic and transcriptomic data from solid and liquid biopsies (pages 15-18). These studies highlight how integrating information from multiple biological sources, along with clinicopathological characteristics, can improve predictive modeling and guide personalized treatment strategies. The updated discussion emphasizes the complementary potential of combining PDOs derived from solid biopsies with CTCs isolated from liquid biopsies for future research and clinical applications.
Reviewer 3 Report
Comments and Suggestions for Authors
The authors provide a review on the multiple potential scientific research uses of patient derived organoids and circulating tumor cells in breast cancer. The text is easy to follow, with a clear presentation of each concept explored by this paper. Some minor grammatical and typographical errors aside, the manuscript makes for a good overwiev of current use of PDOs and CTCs. The only aspect that in my opinion requires improvement is the need for more focus on the breast cancer research performed with these techniques. The very good explanations of the general purpose and use of CTCs and PDOs overshadows the actual application of these techniques in breast cancer. There is a large amount of research conducted in this area, with a subsequent large number of published papers on breast cancer studies with PDOs and CTCs. The authors should highlight these studies more in their review.
Author Response
Response to comments by Reviewer #3:
The authors provide a review on the multiple potential scientific research uses of patient derived organoids and circulating tumor cells in breast cancer. The text is easy to follow, with a clear presentation of each concept explored by this paper. Some minor grammatical and typographical errors aside, the manuscript makes for a good overwiev of current use of PDOs and CTCs. The only aspect that in my opinion requires improvement is the need for more focus on the breast cancer research performed with these techniques. The very good explanations of the general purpose and use of CTCs and PDOs overshadows the actual application of these techniques in breast cancer. There is a large amount of research conducted in this area, with a subsequent large number of published papers on breast cancer studies with PDOs and CTCs. The authors should highlight these studies more in their review.
Our response: We thank the reviewer for the suggestion to strengthen the focus on breast cancer-specific research using PDOs and CTCs. In response, we have revised the manuscript to highlight studies with PDOs and CTCs specifically in breast cancer throughout the relevant sections (pages 5, 6, 7, 12, 13, 15, 16). Additionally, we have added two new tables, which include selected studies using PDOs in breast cancer research (Table 1) and clinical studies using CTCs for prognostic or predictive applications in breast cancer (Table 2).
Round 2
Reviewer 2 Report
Comments and Suggestions for Authors
The quality of the manuscript have been improved a lot, now it can be accept for publishing in our journal.